# Estimating the proportion of beneficial mutations that are not adaptive in mammals

**Thibault Latrille**[1☯]*, **Julien Joseph**[2☯], **Diego A. Hartasánchez**[1], **Nicolas Salamin**[1]

**1** Department of Computational Biology, Université de Lausanne, Lausanne, Switzerland, **2** Laboratoire de Biométrie et Biologie Evolutive, UMR5558, Université Lyon 1, Villeurbanne, France

☯ These authors contributed equally to this work.

\* thibault.latrille@ens-lyon.org

**Data Availability Statement:** The data underlying this article are available at https://doi.org/10.5281/zenodo.7878953. Snakemake pipeline, analysis scripts and documentation are available at https://github.com/ThibaultLatrille/SelCoeff.

## Abstract

Mutations can be beneficial by bringing innovation to their bearer, allowing them to adapt to environmental change. These mutations are typically unpredictable since they respond to an unforeseen change in the environment. However, mutations can also be beneficial because they are simply restoring a state of higher fitness that was lost due to genetic drift in a stable environment. In contrast to adaptive mutations, these beneficial non-adaptive mutations can be predicted if the underlying fitness landscape is stable and known. The contribution of such non-adaptive mutations to molecular evolution has been widely neglected mainly because their detection is very challenging. We have here reconstructed protein-coding gene fitness landscapes shared between mammals, using mutation-selection models and a multi-species alignments across 87 mammals. These fitness landscapes have allowed us to predict the fitness effect of polymorphisms found in 28 mammalian populations. Using methods that quantify selection at the population level, we have confirmed that beneficial non-adaptive mutations are indeed positively selected in extant populations. Our work confirms that deleterious substitutions are accumulating in mammals and are being reverted, generating a balance in which genomes are damaged and restored simultaneously at different loci. We observe that beneficial non-adaptive mutations represent between 15% and 45% of all beneficial mutations in 24 of 28 populations analyzed, suggesting that a substantial part of ongoing positive selection is not driven solely by adaptation to environmental change in mammals.

## Author summary

The extent to which adaptation to changing environments is shaping genomes is a central question in molecular evolution. To quantify the rate of adaptation, population geneticists have typically used signatures of positive selection. However, mutations restoring an ancestral state of higher fitness lost by genetic drift are also positively selected, but they do not respond to a change in the environment. In this study, we have managed to distinguish beneficial mutations that are due to changing environments and those that are

**Funding:** This work was funded by Faculté de Biologie et de Médecine, Université de Lausanne (https://www.unil.ch; to TL, DAH and NS), Swiss National Science Fund (https://www.snf.ch; grant 310030-185223 to NS) and Agence Nationale de la Recherche (https://anr.fr/; grant ANR-19-CE12-0019 / HotRec to JJ). The funders did not play any role in the study design, data collection and analysis, decision to publish, or preparation of the manuscript.

**Competing interests:** The authors have declared that no competing interests exist.

restoring pre-existing functions in mammals. We show that a substantial proportion of beneficial mutations cannot be interpreted as adaptive.

# 1 Introduction

Adaptation is one of the main processes shaping the diversity of forms and functions across the tree of life [1]. Evolutionary adaptation is tightly linked to environmental change and species responding to this change [2, 3]. Such environmental changes are either abiotic (e.g. temperature, humidity) or biotic (e.g. pressure from predators or viruses [4]). For adaptation to occur, there must be variation within populations, which mostly appears via mutations in the DNA sequence. While neutral mutations will not impact an individual fitness, deleterious mutations have a negative effect, and beneficial mutations improve their bearer fitness. A beneficial mutation is thus more likely than a neutral mutation to invade the population and reach fixation, resulting in a substitution at the species level.

Upon environmental change, because adaptive beneficial mutations toward new fitness optima are more likely, the number of substitutions also increases (Fig 1A). An increased substitution rate is thus commonly interpreted as a sign of adaptation [5–7]. The availability of large-scale genomic data and the development of theoretical models have enabled the detection and quantification of substitution rate changes across genes and lineages [8–10]. These approaches, now common practice in evolutionary biology, have helped better understand the processes underpinning the rates of molecular evolution, contributing to disentangling the effects of mutation, selection and drift in evolution [11]. However, a collateral effect has been conflating beneficial mutations with adaptive evolution when adaptive evolution is not the only process that can lead to beneficial mutations [12–14].

## 1.1 Beneficial yet non-adaptive mutations

In a constant environment, a deleterious mutation can reach fixation by genetic drift [15]. A new mutation restoring the ancestral fitness will thus be beneficial (Fig 1B), even though the environment has not changed [13, 16–19]. We will refer to as beneficial non-adaptive mutations those mutations that restore the ancestral fitness under the assumption that the fitness landscape has not changed [12, 20]. Such mutations can happen at a different locus, in which case it is called a compensatory mutation [13, 17]. While compensatory mutations change the sequence and thus induce genetic diversification, beneficial non-adaptive mutations at the locus of the initial mutation reduce genetic diversity and do not contribute to genetic innovation, which are the focus of this manuscript. Although Tomoko Ohta considered beneficial non-adaptive mutations negligible in her nearly-neutral theory [15], their importance has now been acknowledged for expanding populations [12]. However, differentiating between an adaptive mutation and a beneficial non-adaptive mutation remains challenging [21]. Indeed, an adaptive mutation responding to a change in the environment and a beneficial non-adaptive mutation have equivalent fitness consequences for their bearer [12]. Similarly, at the population level, both types of mutations will result in a positive transmission bias of the beneficial allele. However, at the macro-evolutionary scale, the consequences of these two types of mutations are fundamentally different. While adaptive mutations promote phenotype diversification (Fig 1C), beneficial non-adaptive mutations promote phenotype stability and may help preserve well-established biological systems (Fig 1D). Additionally, the direction of adaptive evolution is unpredictable because it is caused by an unforeseen change in the environment and, hence, in the underlying fitness landscape [22]. On the other hand, beneficial non-

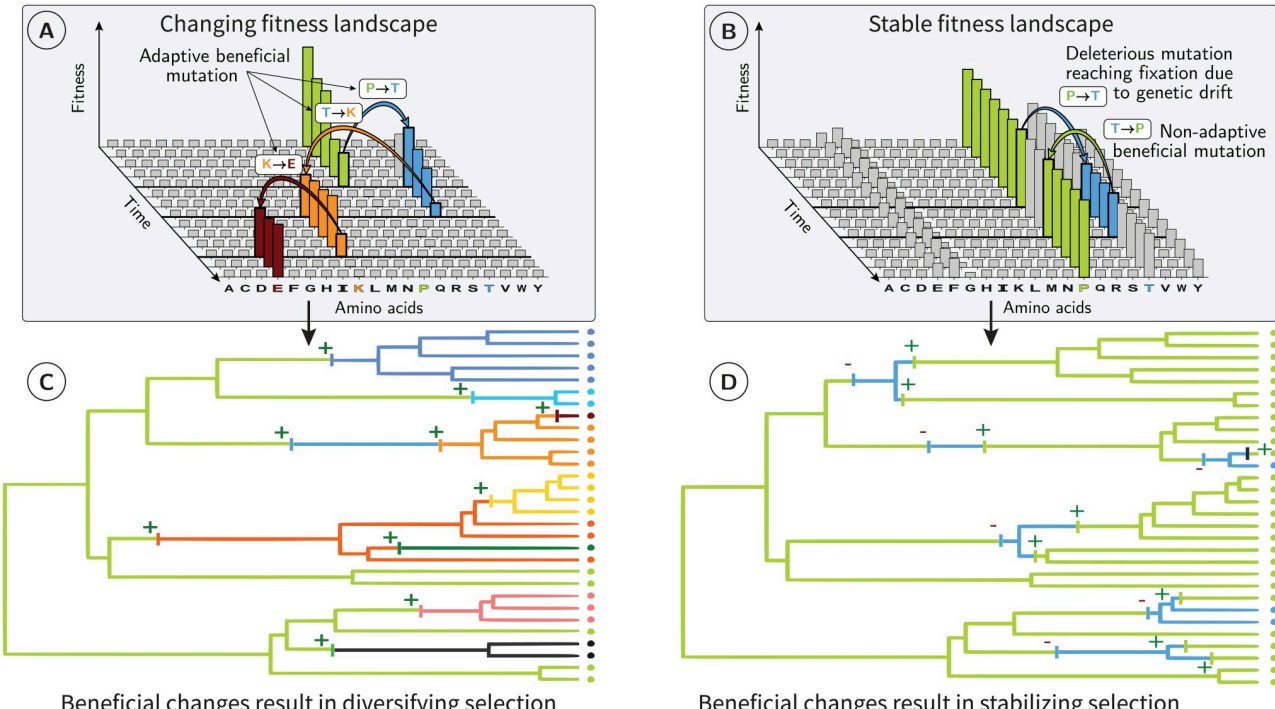

**Fig 1. Changing and stable fitness landscapes.** (A & B) For a given codon position of a protein-coding DNA sequence, amino acids (x-axis) have different fitness values (y-axis). Under a changing fitness landscape (A), these fitnesses fluctuate with time. The protein sequence follows the moving target defined by the amino-acid fitnesses. Since substitutions are preferentially accepted if they are in the direction of this target, substitutions are, on average, adaptive. At the phylogenetic scale (C), beneficial substitutions are common (positive signs), promoting phenotype diversification across species. Under a stable fitness landscape (B), most mutations reaching fixation are either slightly deleterious reaching fixation due to drift or are beneficial non-adaptive mutations restoring a more optimal amino acid. At the phylogenetic scale (D), deleterious substitutions (negative signs) are often reverted via beneficial non-adaptive mutations (positive signs), promoting phenotype stability and preserving well-established biological systems. Even though, individually, any beneficial non-adaptive mutation might have a weak effect on its bearer, we expect them to be scattered across the genome and the genome-wide signature of beneficial non-adaptive mutations to be detectable and quantifiable.

adaptive mutations are predictable because, under a stable fitness landscape, any change from non-optimal to optimal amino acids will move back the site toward the equilibrium expected under the fitness landscape [23–25]. They can then be distinguished from truly novel beneficial mutations because the latter are not expected to mutate toward the amino acids of higher fitnesses defined by the stable fitness landscape but rather mutate to amino acids showing a diversified pattern (Fig 1).

## 1.2 Fitness landscape reconstruction

The mutation-selection framework permits to link the patterns of substitution along a phylogenetic tree with the underlying fitness landscape [26, 27]. Such mutation-selection models applied to protein-coding DNA sequence alignments at the codon level allow us to estimate relative fitnesses for all amino acids for each site of the sequence, explicitly assuming that the underlying fitness landscape is stable along the phylogenetic tree [28–30]. Moreover, effective population size ($N_e$) is considered constant along the phylogenetic tree precisely because of the fixed fitness landscape assumption, the consequences of which are detailed in the Discussion. Importantly, because mutation-selection codon models at the phylogenetic scale are based on population-genetics equations, their estimates of selection coefficients are directly interpretable as fitness effects at the population scale; and because they work at the DNA level, we are

able to account for mutational bias in DNA and structure of the genetic code. The model further integrates the shared evolutionary history between samples and their divergence, which, together, allow us to estimate fitness effects in sequence alignments even though sequences are not independent samples and might not represent the equilibrium distribution of amino acids (see section 4.2 in Materials & methods). The detailed model implementation is available in S1 File, described as a Bayesian hierarchical model (Fig A in S1 File).

Accordingly, fitting the mutation-selection model to a multi-species sequence alignment allows us to obtain relative fitnesses for all amino acids (Fig 2A). The difference in fitness between a pair of amino acids allows us to predict whether any mutation would be a deleterious mutation toward a less fit amino acid, a nearly-neutral mutation, or a mutation toward a known fitter amino acid constituting thus a beneficial non-adaptive mutation (Fig 2B). We can hence use large-scale genomic data to test whether such fitnesses estimated at the phylogenetic scale predict the fitness effects at the population scale. The placental mammals represent an excellent study system to perform such an analysis. Having originated $\sim 102$ million years ago, they diversified quickly [31]. Additionally, polymorphism data are available for many species [32], as are high quality protein-coding DNA alignments across the genome [33, 34]. By performing our analysis on 14,509 orthologous protein-coding genes across 87 species, we focus on genes shared across all mammals in our dataset and not newly functionalized genes in a lineage.

Having identified which potential DNA changes represent beneficial non-adaptive mutations (Fig 2A and 2B), we retrieved polymorphism data from 28 wild and domesticated populations belonging to six genera (*Equus*, *Bos*, *Capra*, *Ovis*, *Chlorocebus*, and *Homo*) to assess the

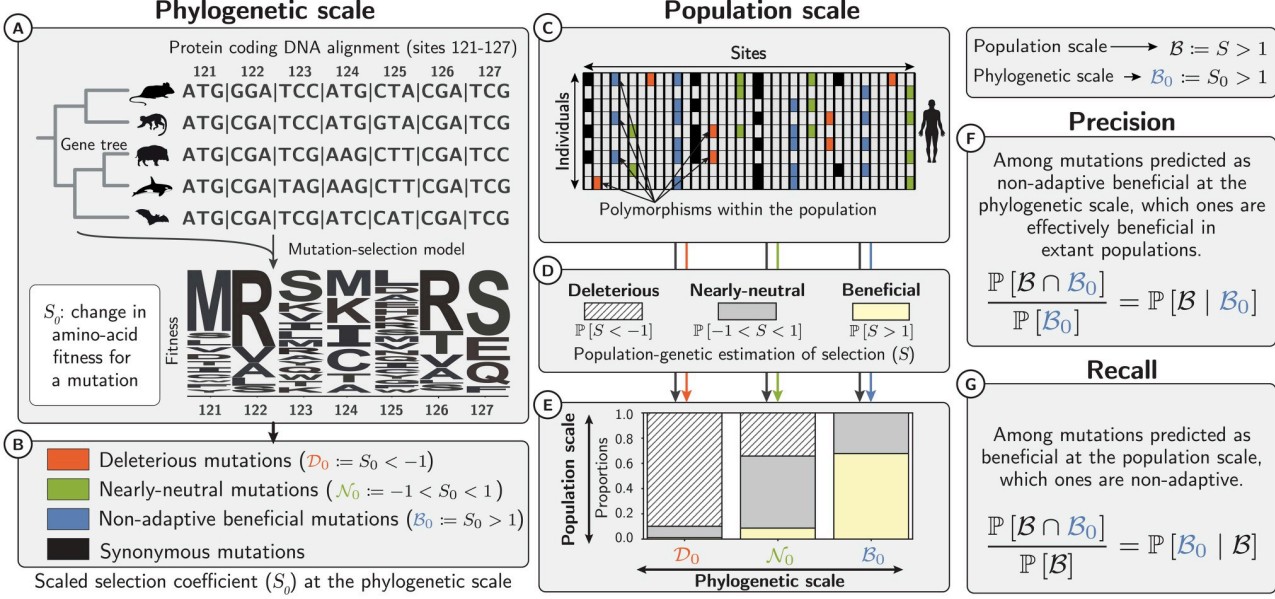

**Fig 2. Selection coefficients at the phylogenetic and population scales.** At the phylogenetic scale (A), we estimated the amino-acid fitness for each site from protein-coding DNA alignments using mutation-selection codon models. For every possible mutation, the difference in amino-acid fitness before and after the mutation allows us to compute the selection coefficient at the phylogenetic scale ($S_0$). Depending on $S_0$ (B), mutations can be predicted as deleterious ($\mathcal{D}_0$), nearly-neutral ($\mathcal{N}_0$) or beneficial non-adaptive mutations ($\mathcal{B}_0$) toward a fitter amino acid and repairing existing functions. At the population scale, each observed single nucleotide polymorphism (SNP) segregating in the population can also be classified according to its $S_0$ value (C). Occurrence and frequency in the population of non-synonymous polymorphisms, contrasted to synonymous polymorphisms (deemed neutral), is used to estimate selection coefficients (D-E) at the population scale ($S$), for each class of selection ($\mathcal{D}_0$, $\mathcal{N}_0$, $\mathcal{B}_0$). We can thus assess whether $S_0$ predicts $S$ and compute *precision* (F) and *recall* (G) for each class. The *recall* value for class $\mathcal{B}_0$ is the probability for beneficial mutations to be non-adaptive (G). Icons are adapted from https://phylopic.org under a Creative Commons license.

presence of beneficial non-adaptive mutations at the population scale. We focused on both mutations currently segregating within populations and on substitutions in the terminal branches, and checked if any of these observed changes were indeed beneficial (Fig 2C and 2E). A similar approach demonstrated the presence of beneficial non-adaptive mutations in humans [23, 24] and in plants [25]. However, the model used to reconstruct the static fitness landscape in these studies can only be applied to deeply conserved protein domains in the tree of life, which corresponds to a subpart of the proteome that evolves slowly. The mutation-selection model used in the present work integrates phylogenetic relationships, and thus allows us to estimate the fitness landscape in shallower phylogenetic trees, and therefore can be applied almost exome-wide [35].

We first quantified the likelihood of any DNA mutation to be a beneficial non-adaptive mutation, that is, whenever a DNA mutation increases fitness under a stable fitness landscape. Subsequently, by quantifying the total amount of beneficial mutations in the current population across all types of DNA mutations, we could tease apart beneficial non-adaptive from adaptive mutations resulting from a change in the fitness landscape. Altogether, in this study, by integrating large-scale genomic datasets at both phylogenetic and population scales, we propose a way to explicitly quantify the contribution of beneficial non-adaptive mutations to positive selection across the entire exome of the six genera (Fig 2F and 2G).

## 2 Results

### 2.1 Selection along the terminal branches

First, we assessed whether fitness effects derived from the mutation-selection model at the phylogenetic scale predict selection occurring in terminal branches. We recovered the mutations that reached fixation in the terminal branches of the six genera. We only considered mutations fixed in a population as substitutions in the corresponding branch by discarding mutations segregating in our population samples. For each substitution identified in the terminal branches we obtained its $S_0$ value such as predicted at the mammalian scale (Fig 2A and 2B). We could classify each substitution as either deleterious ($\mathcal{D}_0 := S_0 < -1$), nearly-neutral ($\mathcal{N}_0 := -1 < S_0 < 1$), or beneficial ($\mathcal{B}_0 := S_0 > 1$). Because $S_0$ values were based on the assumption that the fitness landscape is stable across mammals, $\mathcal{B}_0$ mutations (i.e., with $S_0 > 1$) bring the bearer of this mutation toward an amino acid predicted to be fitter across mammals. Importantly, the mammalian alignment used to estimate the amino acid fitness landscape did not include the six focal genera and their sister species. This ensures independence between, on the one hand, the fitness landscape estimated, and on the other hand, both substitutions that occurred in the terminal branches, and segregating polymorphisms of the focal populations. Example substitutions in the terminal lineage of *Chlorocebus sabaeus* which are classified as $\mathcal{B}_0$ are shown in S2 File (section 1.1). For instance, in the mammalian protein-coding DNA alignment of gene *SELE*, the nucleotide at site 1722 has mutated (from T to C) at the base of Simiiformes (monkeys and apes), modifying the corresponding amino acid from Serine to Proline, but has been subsequently reverted in the branch of *Chlorocebus sabaeus* (Fig A in S2 File). However, other substitutions classified as $\mathcal{B}_0$ in the terminal branch of *Chlorocebus sabaeus* cannot be clearly interpreted as reversions along the terminal branch, and show several transitions to this amino acid across the mammalian phylogeny, as for instance site 3145 of gene *THSD7A* (Fig B in S2 File).

Among all the substitutions found in each terminal branch, between 10 and 13% were $\mathcal{B}_0$, while $\mathcal{B}_0$ mutations only represent between 0.9 and 1.2% of all non-synonymous mutations (Fig 3A and 3B for humans, Table A in S2 File for all dataset). Of note, if we were to assume a stationary mutation-selection-drift equilibrium in the terminal lineage, we would expect a

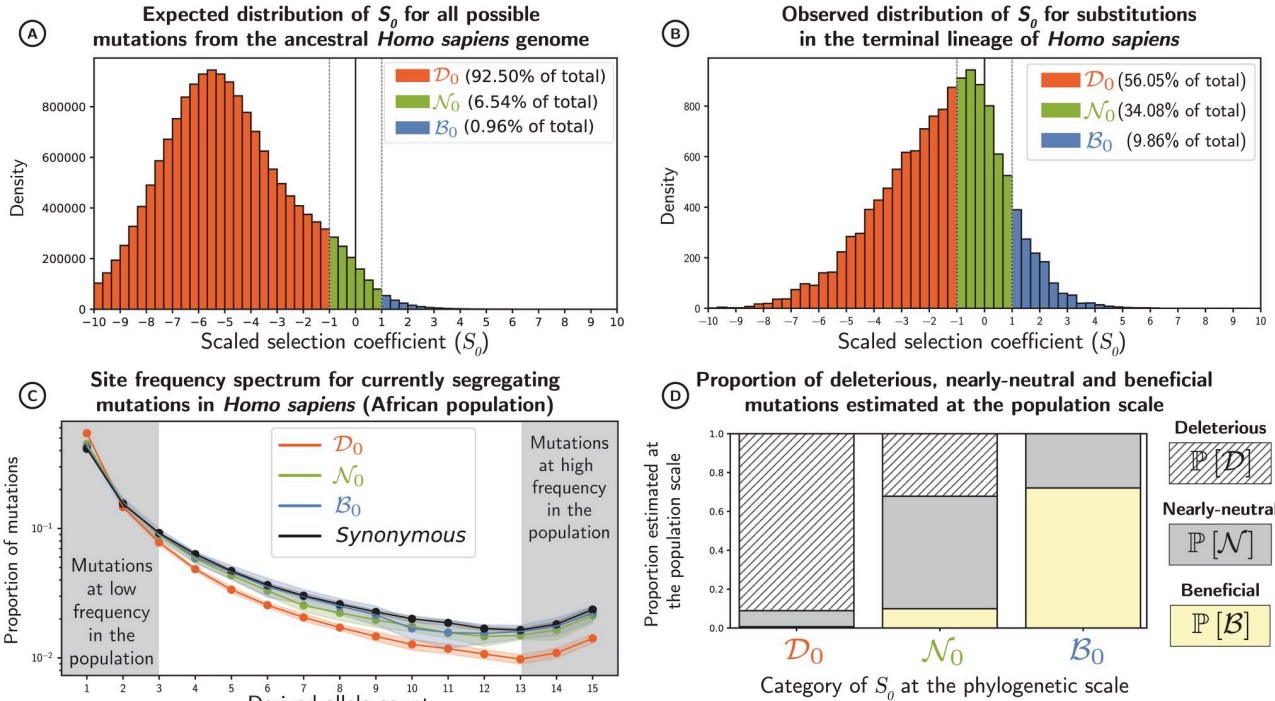

**Fig 3. Selection coefficients of mutations in humans of African descent.** (A) Distribution of scaled selection coefficients ($S_0$), predicted for all possible non-synonymous DNA mutations away from the ancestral human exome (section 4.4). Mutations are divided into three classes of selection: deleterious ($\mathcal{D}_0$), nearly-neutral ($\mathcal{N}_0$) and beneficial ($\mathcal{B}_0$, supposedly beneficial non-adaptive mutations) (B) Distribution of scaled selection coefficients ($S_0$) for all observed substitutions along the *Homo* branch after the *Homo-Pan* split (section 4.5). If there are fewer substitutions than expected, this class is thus undergoing purifying selection, as is the case for $\mathcal{D}_0$. (C) The site-frequency spectrum (SFS) in humans of African descent for a random sample of 16 alleles (means in solid lines and standard deviations in color shades) for each class of selection and for synonymous mutations, supposedly neutral (black). The SFS represents the proportion of mutations (y-axis) with a given number of derived alleles in the population (x-axis). At high frequencies, deleterious mutations are underrepresented. (D) Proportion of beneficial $\mathbb{P}[\mathcal{D}]$, nearly-neutral $\mathbb{P}[\mathcal{N}]$, and deleterious mutations $\mathbb{P}[\mathcal{B}]$ estimated at the population scale for each class of selection at the phylogenetic scale (section 4.6). Proportions depicted here are not weighted by their mutational opportunities.

symmetric proportion of positively ($\mathcal{B}_0$) and negatively ($\mathcal{D}_0$) selected substitutions if there were no adaptation. The lack of symmetry along the terminal branches then provides a means to estimate the frequency of non-adaptive beneficial substitutions. Mathematically, twice the fraction of $\mathcal{B}_0$ substitutions is an estimate of this rate. This rate is highly consistent across lineages (Table A in S2 File) and suggests an overall frequency of nearly neutral substitutions due to consistent long-term selection pressures, mutation and drift of approximately 20% of all substitutions (20%-26% across species).

Furthermore, since in principle, $\mathcal{B}_0$ mutations are bound to reach fixation more often than neutral mutations, we calculated the $d_N/d_S$ ratio of non-synonymous over synonymous divergence for all terminal lineages, focusing on the non-synonymous changes predicted as $\mathcal{B}_0$ mutations ($d_N(\mathcal{B}_0)/d_S$). We obtained values between 1.17 and 1.75 in the different lineages (Table B in S2 File), meaning that $\mathcal{B}_0$ mutations reach fixation slightly more frequently than synonymous mutations that are supposed to be neutral, consistent with these $\mathcal{B}_0$ mutations being weakly beneficial. Such an observation is consistent with the premise that $\mathcal{B}_0$ mutations are weakly beneficial, translating to a scaled selection coefficient between 0.32 and 1.24 (section 1.2 in S2 File). Finding $d_N(\mathcal{B}_0)/d_S > 1$ for these sites confirms that these sites are closer to optimality at the end of the branch than at the beginning. Even though the beneficial effect of

these mutations does not come from an environmental change, it does not change the fact that they have contributed positively to the population's fitness. It is just that at mutation-selection-drift equilibrium, the increase in fitness at these sites is offset by deleterious substitutions elsewhere in the genome so that there is no net adaptation.

This result further indicates that using $d_N/d_S$ as an estimate of purifying selection is biased (overestimated) due to the presence of beneficial non-adaptive mutations among the non-synonymous substitutions. By discarding all beneficial non-adaptive mutations we can obtain an estimate of $d_N/d_S$ which is not inflated. By comparing these two ways of calculating $d_N/d_S$ (see section 4.5 in Materials & methods), we calculated that beneficial non-adaptive mutations inflate $d_N/d_S$ values by between 9 and 12% across genera (Table C in S2 File). This represents a substantial increase when considering that beneficial non-adaptive mutations only represent between 0.9 and 1.2% of non-synonymous mutational opportunities (Table A in S2 File).

## 2.2 Selection in populations

Second, we assessed whether our calculated $S_0$ values predicted at the phylogenetic scale were also indicative of the selective forces exerted at the population level. We retrieved single nucleotide polymorphisms (SNPs) segregating in 28 mammalian populations. To determine if SNPs were ancestral or derived, we reconstructed the ancestral exome of each population. We then classified every non-synonymous SNP as either $\mathcal{D}_0$, $\mathcal{N}_0$, or $\mathcal{B}_0$ according its $S_0$ value (Fig 2B and 2C).

First, SNPs classified as $\mathcal{B}_0$ are spread across the genomes and not strongly associated to the ontology terms of their respective genes (Table D and Fig C in S2 File). In humans, some SNPs have been associated with specific clinical prognosis terms obtained by clinical evaluation of the impact of variants on human Mendelian disorders [36]. Although this classification also relies on deep protein alignments and therefore cannot be considered an independent result from our own, it does provide a consistency check if the effect of a mutation on human health is in line with its fitness effect predicted by our method [37]. Therefore, we investigated whether the non-synonymous SNPs classified as $\mathcal{D}_0$ or $\mathcal{B}_0$ showed enrichment in specific clinical terms compared to SNPs classified as $\mathcal{N}_0$. Our results show that SNPs predicted as deleterious are associated with clinical terms such as *Likely Pathogenic* and *Pathogenic*, implying that, in general, the selective pressure of a mutation exerted across mammals is also predictive of its clinical effect in humans (Table E in S2 File) [38]. Conversely, $\mathcal{B}_0$ mutations are associated with clinical terms such as *Benign* and *Likely Benign*, which shows that $\mathcal{B}_0$ mutations are less likely to be functionally damaging (Table F in S2 File).

In addition to clinical prognosis, frequencies at which SNPs are segregating within populations provide information on their selective effects. For instance, deleterious SNPs usually segregate at lower frequencies because of purifying selection, which tends to remove them from the population (Fig 3C for humans). By gathering information across many SNPs, it is possible to estimate the distribution of fitness effects at the population scale, taking synonymous SNPs as a neutral expectation [39–42]. From these estimated fitness effects, we can derive the proportion of deleterious mutations ($\mathbb{P}[\mathcal{D}]$), nearly-neutral mutations ($\mathbb{P}[\mathcal{N}]$) and beneficial mutations ($\mathbb{P}[\mathcal{B}]$) at the population scale (see section 4.6 in Materials & methods, Fig A-C in S3 File). These approaches offer a unique opportunity to contrast selection coefficients estimated at the phylogenetic scale ($S_0$) and at the population scale ($S$) in different dataset (Fig D in S3 File).

Across our selection classes ($\mathcal{D}_0$, $\mathcal{N}_0$ and $\mathcal{B}_0$), one can ultimately estimate the proportion of correct and incorrect predictions, leading to an estimation of *precision* and *recall* (Fig 2F and 2G and section 4.7 in Materials & methods). Across 28 populations of different mammal

species, mutations predicted to be deleterious at the phylogenetic scale ($\mathcal{D}_0$) were indeed purged at the population scale, with a *precision* in the range of 90–97% (Table 1 and Fig 3D for humans). Conversely, a *recall* in the range of 96–100% implied that mutations found to be deleterious at the population scale were most likely also predicted to be deleterious at the phylogenetic scale (Table 1). Altogether, purifying selection is largely predictable and amino acids with negative fitness across mammals have been effectively purged away in each population.

Mutations predicted as $\mathcal{N}_0$ were effectively composed of a mix of neutral and selected mutations with varying *precision* (36–63%) and *recall* (32–45%) across the different populations (Table 1, Fig 3D for humans). The variable proportions between populations can be explained by the effective number of individuals in the population ($N_e$), a major driver of

**Table 1. *Precision* and *recall* for estimated selection coefficient of mutations given by mutation-selection models ($S_0$).**

| Population | Species | $N_e$ | Deleterious mutations $\mathcal{D} := S < -1$ $\mathcal{D}_0 := S_0 < -1$ | | Nearly-neutral mutations $\mathcal{N} := -1 < S < 1$ $\mathcal{N}_0 := -1 < S_0 < 1$ | | Beneficial mutations $\mathcal{B} := S > 1$ $\mathcal{B}_0 := S_0 > 1$ | |
|---|---|---|---|---|---|---|---|---|
| | | | Precision $\mathbb{P}[\mathcal{D} \mid \mathcal{D}_0]$ | Recall $\mathbb{P}[\mathcal{D}_0 \mid \mathcal{D}]$ | Precision $\mathbb{P}[\mathcal{N} \mid \mathcal{N}_0]$ | Recall $\mathbb{P}[\mathcal{N}_0 \mid \mathcal{N}]$ | Precision $\mathbb{P}[\mathcal{B} \mid \mathcal{B}_0]$ | Recall $\mathbb{P}[\mathcal{B}_0 \mid \mathcal{B}]$ |
| Equus c. | Equus caballus | $7.5 \times 10^4$ | 0.923 | 0.972 | 0.570 | 0.341 | 0.648 | 0.536 |
| Iran | Bos taurus | $5.6 \times 10^4$ | 0.915 | 1.000 | 0.632 | 0.358 | 0.873 | 0.243 |
| Uganda | Bos taurus | $1.3 \times 10^5$ | 0.951 | 0.969 | 0.495 | 0.414 | 0.576 | 0.415 |
| Australia | Capra hircus | $1.7 \times 10^5$ | 0.944 | 0.971 | 0.527 | 0.437 | 0.368 | 0.177 |
| France | Capra hircus | $1.9 \times 10^5$ | 0.946 | 0.971 | 0.508 | 0.423 | 0.368 | 0.190 |
| Iran (C. aegagrus) | Capra hircus | $1.9 \times 10^5$ | 0.948 | 0.969 | 0.486 | 0.444 | 0.368 | 0.165 |
| Iran | Capra hircus | $2.3 \times 10^5$ | 0.953 | 0.966 | 0.425 | 0.407 | 0.368 | 0.193 |
| Italy | Capra hircus | $1.9 \times 10^5$ | 0.947 | 0.971 | 0.551 | 0.439 | 0.368 | 0.243 |
| Morocco | Capra hircus | $2.2 \times 10^5$ | 0.950 | 0.970 | 0.527 | 0.440 | 0.368 | 0.245 |
| Iran | Ovis aries | $3.8 \times 10^5$ | 0.961 | 0.961 | 0.452 | 0.415 | 0.205 | 0.407 |
| Iran (O. orientalis) | Ovis aries | $4.5 \times 10^5$ | 0.964 | 0.960 | 0.420 | 0.445 | 0.193 | 0.190 |
| Iran (O. vignei) | Ovis aries | $3.7 \times 10^5$ | 0.967 | 0.959 | 0.361 | 0.470 | 0.190 | 0.110 |
| Various | Ovis aries | $4.1 \times 10^5$ | 0.962 | 0.962 | 0.433 | 0.440 | 0.229 | 0.222 |
| Morocco | Ovis aries | $4 \times 10^5$ | 0.962 | 0.961 | 0.462 | 0.424 | 0.211 | 0.514 |
| Barbados | Chlorocebus sabaeus | $1.1 \times 10^5$ | 0.935 | 0.975 | 0.565 | 0.402 | 0.648 | 0.293 |
| Central Afr. Rep. | Chlorocebus sabaeus | $1.7 \times 10^5$ | 0.948 | 0.971 | 0.508 | 0.423 | 0.535 | 0.275 |
| Ethiopia | Chlorocebus sabaeus | $1.4 \times 10^5$ | 0.935 | 0.975 | 0.580 | 0.416 | 0.552 | 0.245 |
| Gambia | Chlorocebus sabaeus | $1.4 \times 10^5$ | 0.944 | 0.975 | 0.654 | 0.437 | 0.577 | 0.821 |
| Kenya | Chlorocebus sabaeus | $1.5 \times 10^5$ | 0.946 | 0.972 | 0.538 | 0.453 | 0.588 | 0.257 |
| Nevis | Chlorocebus sabaeus | $1 \times 10^5$ | 0.933 | 0.976 | 0.629 | 0.412 | 0.599 | 0.358 |
| South Africa | Chlorocebus sabaeus | $1.8 \times 10^5$ | 0.944 | 0.971 | 0.548 | 0.423 | 0.574 | 0.341 |
| Saint Kitts | Chlorocebus sabaeus | $1.2 \times 10^5$ | 0.936 | 0.975 | 0.586 | 0.402 | 0.598 | 0.336 |
| Zambia | Chlorocebus sabaeus | $1.7 \times 10^5$ | 0.945 | 0.971 | 0.512 | 0.432 | 0.585 | 0.250 |
| African | Homo sapiens | $5.6 \times 10^4$ | 0.911 | 0.976 | 0.579 | 0.325 | 0.721 | 0.349 |
| Admixed American | Homo sapiens | $4.5 \times 10^4$ | 0.902 | 0.978 | 0.584 | 0.299 | 0.690 | 0.345 |
| East Asian | Homo sapiens | $4 \times 10^4$ | 0.905 | 0.978 | 0.585 | 0.325 | 0.688 | 0.249 |
| European | Homo sapiens | $4.2 \times 10^4$ | 0.906 | 0.978 | 0.584 | 0.329 | 0.688 | 0.248 |
| South Asian | Homo sapiens | $4.4 \times 10^4$ | 0.908 | 0.978 | 0.584 | 0.342 | 0.691 | 0.224 |

*Precision* is the estimation of the selection coefficient at population scale ($S$) given that $S_0$ is known. Conversely, *recall* is the estimation of $S_0$ given selection coefficient at the population scale ($S$) is known. *Recall* for beneficial mutations ($\mathbb{P}[\mathcal{B}_0 \mid \mathcal{B}]$) is thus the proportion of beneficial non-adaptive mutations among all beneficial mutations. $N_e$ is the estimated effective population size for each population.

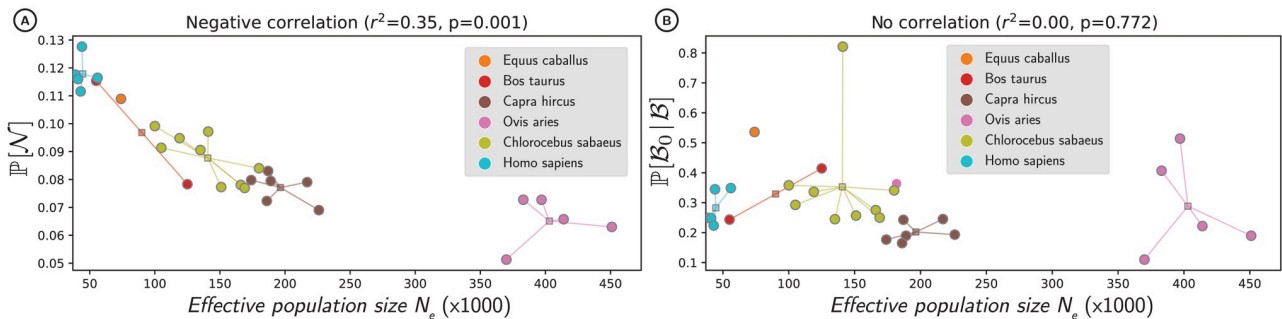

**Fig 4. Proportion of nearly-neutral mutations and beneficial non-adaptive mutations as a function of effective population size ($N_e$).** Populations in circles, mean of the species across the populations as squares. (A) Proportion of nearly-neutral mutations at the population scale ($\mathbb{P}[\mathcal{N}]$ in the y-axis), shown as a function of estimated effective population size ($N_e$ in the x-axis). (B) Proportion of beneficial non-adaptive mutations among all beneficial mutations ($\mathbb{P}[\mathcal{B}_0 \mid \mathcal{B}]$ in the y-axis), shown as a function of $N_e$ in x-axis. Correlations account for phylogenetic relationship and non-independence of samples, through the fit of a Phylogenetic Generalized Linear Model (see section 4.6 in Materials & methods).

selection efficacy. Moreover, estimates of mutation rate per generation ($u$), from Bergeron *et al.* [43] and Orlando *et al.* [44], and Watterson's $\theta$ obtained from the synonymous SFS as in Achaz [45], allow us to obtain $N_e$ through $N_e = \theta/4u$. Using correlation analyses that accounted for phylogenetic relationship (see section 4.8 in Materials & methods, Fig E in S3 File), we found that higher $N_e$ was associated with a smaller proportion of nearly-neutral mutations ($r^2 = 0.31$, $p = 0.001$, Fig 4A). This result follows the prediction of the nearly-neutral theory and suggests that in populations with higher diversity (e.g., *Bos* or *Ovis*), discrimination between beneficial and deleterious mutations is more likely to occur (Fig F-H in S3 File). Conversely, many more mutations are effectively neutral in populations with lower diversity (e.g., *Homo*).

Finally, mutations predicted to be $\mathcal{B}_0$ were indeed beneficial for individuals bearing them, with a *precision* (Fig 2F) in the range of 19–87% (Table 1 and Fig 3D for humans). This result confirms that selection toward amino acids restoring existing functions is ongoing in these populations. Importantly, the *recall* value in this case, computed as $\mathbb{P}[\mathcal{B}_0 \mid \mathcal{B}]$, is the probability for a beneficial mutation at the population scale to be a non-adaptive, i.e., going toward a fitter amino acid given a stable fitness landscape (Fig 2G, Table A in S4 File). In other words, the *recall* value quantifies the number of beneficial mutations restoring damaged genomes instead of creating adaptive innovations. Across the 28 populations, this proportion is in the range of 11–82% (Table 1), with a mean of 30%. Accounting for phylogenetic relationships, we found no correlation between the proportion of beneficial non-adaptive mutations and estimates of $N_e$ based on genetic diversity ($r^2 = 0.00$, $p = 0.772$, Fig 4B).

We additionally performed controls and simulations to ensure that our results were robust. First, we controlled that these estimations were not affected by SNP mispolarization (Fig A-B in S4 File). Second, we performed simulations at the population-genetic level and confirmed that our method was able to recover the proportion of beneficial mutations that are non-adaptive in synthetic polymorphism datasets (Fig C in S4 File). Third, we ran our analysis filtering out CpG mutations and obtained values of $\mathbb{P}[\mathcal{B}_0 \mid \mathcal{B}]$ in the range of 5–27%, with a mean of 14% (Table B-C in S4 File, Fig D in S4 File), providing more conservative estimates. Finally, because the phylogenetic mutation-selection codon model should fit better for genes with uniformly conserved functions, we filtered out genes under pervasive adaptation [46] as a control. In this subset of the exome, containing genes with a more stable fitness landscape, we found an increase in the proportion of beneficial mutations that are non-adaptive (Wilcoxon signed-

rank, $s = 80$, $p = 0.002$, Table D in S4 File), consistent with our expectation that beneficial mutations occur more frequently in genes under changing fitness landscapes.

## 2.3 Selection in the terminal lineage and in populations

As an alternative to relying solely on currently segregating mutations to quantify selection, one can leverage both polymorphism within a population and substitutions in the terminal lineage to estimate the distribution of fitness effects (DFE). Hence, we estimated *precision* and *recall* as done previously, but now including the number of substitutions per site as input for the DFE estimators (see 4.6 in Materials & methods and Fig E in S4 File). When including substitutions in the terminal lineage, estimates of $\mathbb{P}[\mathcal{B}_0 \mid \mathcal{B}]$ are in the 10–78% range with a mean of 36%, and 19 out of 28 estimates fall between 15% and 45% (Table E-F in S4 File).

Additionally, we controlled that these estimations were not affected by SNP mispolarization (Fig F in S4 File). We also filtered out genes under pervasive adaptation, and again found an increase in $\mathbb{P}[\mathcal{B}_0 \mid \mathcal{B}]$, consistent with our expectation (Wilcoxon signed-rank, $s = 120$, $p = 0.027$, Table G in S4 File). We assessed the impact of fitting the same functional form of DFE to the three different categories of changes $\mathcal{D}_0$, $\mathcal{N}_0$ and $\mathcal{B}_0$. To this aim we computed the total amount of current selection by fitting either a single DFE on the whole dataset or by summing the other three independent DFEs. These disjoint estimates are well correlated, with a goodness of fit $r^2 = 0.95, 0.89, 0.82$ for respectively $\mathbb{P}[\mathcal{D}]$ (Fig G in S4 File), $\mathbb{P}[\mathcal{N}]$ (Fig H in S4 File) and $\mathbb{P}[\mathcal{B}]$ (Fig I in S4 File). Finally, we evaluated the effect of fitting a parametric functional form for the DFE. As implemented in Tataru *et al.* [42], the DFE is a mixture between a reflected gamma distribution and an exponential distribution (Eq 8, section 4.7 in Materials & methods). Instead of using such a continuous DFE, we also tested our prediction with a non-parametric functional form for the DFE, obtaining estimates of $\mathbb{P}[\mathcal{B}_0 \mid \mathcal{B}]$ in the 8–94% range, with a mean of 43% (Table H-I in S4 File).

## 3 Discussion

### 3.1 Beneficial mutations are not necessarily adaptive

This study represents an essential step toward integrating the different evolutionary scales necessary to understand the combined effects of mutation, selection, and drift on genome evolution. In particular, we have been able to quantify the proportion of beneficial mutations that are non-adapative (i.e., not a response to a change in fitness landscape), which has only been achievable by combining exome-wide data from both phylogenetic and population scales. At the phylogenetic scale, codon diversity at each site of a protein-coding DNA alignment allows for reconstructing an amino-acid fitness landscape, assuming that this landscape is stable along the phylogenetic tree. These amino-acid fitness landscapes allow us to predict any mutation's selection coefficient ($S_0$) along a protein-coding sequence. We have compared these selective effects to observations at the population level, and by doing so, we have confirmed that mutations predicted to be deleterious ($\mathcal{D}_0 \coloneqq S_0 < -1$) are generally purified away in extant populations. Our results concur with previous studies showing that SIFT scores [47, 48], based on amino acid alignments across species, also inform on the deleterious fitness effects exerted at the population scale [25]. However, contrary to SIFT scores, our mutation-selection model is parameterized by a fitness function such that changes are directly interpretable as fitness effects (see also section 1 in S3 File). In this regard, an interesting prediction of our model is that some deleterious mutations reach fixation due to genetic drift, while beneficial non-adaptive mutations restore states of higher fitness. We have tested this hypothesis and have found that a substantial part of these predicted non-adaptive mutations ($\mathcal{B}_0 \coloneqq S_0 > 1$) are

indeed beneficial in extant populations. We estimate that between 11 and 82% of all beneficial mutations in mammalian populations are not adaptive. More specifically, in 24 out of 28 populations analyzed, the percentage of beneficial mutations estimated to be non-adaptive falls between 15 and 45%. These results suggest that many beneficial mutations are not adaptive, but rather restore states of higher fitness. Hence, we can correctly estimate the extent of adaptive evolution only if we account for the number of beneficial non-adaptive mutations [49, 50]. Here instead, we argue that we should dissociate positive selection from adaptive evolution and limit the use of adaptive mutations to those that are associated with adaptation to environmental change as such [12, 13, 51].

## 3.2 Assumptions and methodological limitations

The exact estimation of the contribution of beneficial non-adaptive mutations to positive selection relies on some hypotheses at both the phylogenetic and population scales and is sensitive to methodological limitations. Indeed, data quality and potentially inadequate modeling choices of both the fitness landscape (at the phylogenetic scale) and fitness effects (at the population scale) might also lead to missed predictions [10]. In practice, we obtained different values of the proportion of non-adaptive beneficial mutations depending on i) the filtering or not of CpG mutations [52], ii) whether we included substitutions in the terminal lineage along with within-population polymorphisms to estimate fitness effects [42], and iii) the model used to infer the fitness effects. It appears that our estimation can be sensitive to model misspecification and overall, while we provide an order of magnitude for the contribution of beneficial non-adaptive mutations to positive selection, methodological improvement on the estimation of the DFE is needed to increase the precision of this value.

To be conservative, we considered mutations as adaptive if they were detected as being under positive selection at the population scale despite them being either incorrectly predicted as deleterious ($\mathcal{D}_0$) or nearly-neutral ($\mathcal{N}_0 := -1 < S_0 < 1$) from the amino-acid mammalian fitnesses. An example of an incorrectly predicted deleterious mutation ($\mathcal{D}_0$) from its fitness landscape could be an amino acid having always been deleterious across mammals, but being advantageous ($\mathcal{B}$) in the current species due to environmental changes or a major shift in their fitness landscape (e.g. domestication). To visualize an example of a wrongly predicted nearly-neutral mutation, we can first imagine a site where only hydrophilic amino acids are accepted because of the protein properties (e.g. a surface site of a globular protein). Let us then assume that such a site is also a target for viruses, hence promoting amino-acid changes which modify the site's viral affinity [4]. Given the selective pressure favoring amino-acid change, but restricting the possibilities to hydrophilic amino acids, most hydrophilic amino acids will likely be visited along the phylogenetic tree and the mutation-selection model will give high and similar fitnesses to all of them. In such a case, any mutation between hydrophilic amino acids will be wrongly predicted as nearly-neutral ($\mathcal{N}_0$), while it is in fact adaptive. In summary, under a changing fitness landscapes [53], our phylogenetic mutation-selection model takes an average over fitness changes observed along the phylogeny, causing beneficial mutations ($\mathcal{B}$) to be predicted as either deleterious ($\mathcal{D}_0$) or nearly-neutral ($\mathcal{N}_0$), therefore mechanically reducing $\mathbb{P}[\mathcal{B}_0 \mid \mathcal{B}]$, and making our estimate conservative.

## 3.3 Convergent adaptation

If there are several substitutions toward the same amino acid along the mammalian tree (section 1.1 in S2 File), our mutation-selection model cannot formally distinguish between a scenario where mammals have fixed deleterious mutations that are reverted in several lineages, from more complex scenarios involving convergent adaptation across mammals. In a first

scenario, repeated changes of fitness landscapes in the same direction could occur along several lineages, leading to repeated substitutions in multiple lineages (parallel or convergent adaptation). In a second scenario, an environmental change that occurred near the root of placental mammals ($\sim$100 Mys ago), to which extant populations are currently responding independently through weakly adaptive mutations, could also lead to repeated substitutions toward the same amino acids. Importantly, we would usually expect adaptive convergent mutations to be linked to particular converging phenotypes across mammals, and hence, they should not massively affect the whole genome as we find (Fig C and Table D in S2 File). Moreover, after filtering out genes usually associated to recurrent adaptation (e.g. immune genes), we recover an even higher proportion of beneficial non-adaptive mutations (Table D in S4 File). For these reasons, we argue that the signal of predictable positive selection we recover in extant population is indeed mainly driven by non-adaptive evolution.

## 3.4 The influence of effective population size

Across the genome, beneficial non-adaptive mutations and deleterious mutations reaching fixation create a balance in which genomes are constantly damaged and restored simultaneously at different loci due to drift. Since the probability of fixation of mutations depends on the effective population size ($N_e$), the history of $N_e$ plays a crucial role in determining the number of beneficial non-adaptive mutations compensating for deleterious mutations [54]. For example, a population size expansion will increase the efficacy of selection, and a larger proportion of mutations will be beneficial (otherwise effectively neutral), thus increasing the number of beneficial non-adaptive mutations. On the other hand, a population that has experienced a high $N_e$ throughout its history should be closer to an optimal state under a stable fitness landscape, having suffered fewer fixations of deleterious mutations and therefore decreasing the probability of beneficial non-adaptive mutations [55]. Overall, we expect the proportion of beneficial non-adaptive mutations to be more dependent on $N_e$'s long-term expansions and contractions than on the short-term ones [12, 55].

Moreover, because our model assumes a fixed fitness landscape, it implicitly assumes that $N_e$ is constant along the phylogenetic tree. Fluctuations due to changes in the fitness landscape or in $N_e$ will be averaged out by the assumption of the current model that $N_e$ is constant across lineages. It was recently shown [54], using computer intensive mutation-selection models with fluctuating $N_e$, that relaxing the assumption of a constant $N_e$ results in more extreme estimates of amino-acid fitnesses than with the standard model used in this study. In other words, by assuming a constant $N_e$, we are underpowered to detect beneficial non-adaptive mutations since amino acids will have more similar fitnesses. As a consequence, some of the beneficial non-adaptive mutations currently segregating in population will be incorrectly classified as nearly-neutral by the mutation-selection model, and thus be wrongly interpreted as adaptive (see previous section). This ultimately results in lower estimates of the proportion of beneficial non-adaptive mutations. Given this inflation of missed predictions due to change in population sizes [14, 56, 57], our estimated proportion of beneficial non-adaptive mutations among adaptive ones is likely to be an underestimation.

## 3.5 The role of epistasis and compensatory mutations

Our model assumes that amino-acid fitness landscapes are site-specific and also independent of one another, whereas under pervasive epistasis, the fitness effect of any mutation at a particular site would depend on the amino acids present at other sites. Epistasis is common for mutations that influence the protein's physical properties (e.g. conformation, stability, or affinity for ligands) or might arise due to nonlinear relationship between the protein's physical

properties and fitness [58]. Regardless of its origin, epistasis has been shown to play a role in the evolution of protein-coding genes, with amino-acid residues in contact within a protein or between proteins tending to co-evolve [58–60]. Particularly, the residues in contact co-evolve to become more compatible with each other generating an entrenchment [61–63]. Epistasis therefore allows for compensatory mutations, which restore fitness through mutations at loci different from where deleterious mutations took place, representing another case of non-adaptive beneficial mutations, but one which is not accounted for by our method. Hence, the beneficial mutations that we classify as putatively adaptive might in fact be compensatory mutations, making our estimation of the rate of non-adaptive beneficial mutations conservative.

Despite epistasis being an important factor in protein evolution, several deep-mutational scanning experiments have revealed that a site-specific fitness landscape predicts the evolution of sequences in nature with considerable accuracy [64–66]. Additionally, the fact that we observe such a high proportion of beneficial non-adaptive mutations suggests that the underlying assumptions of our model, namely site-independence, implying no epistasis, and a static fitness landscape, are a reasonable approximation for the underlying fitness landscape of proteins. Our results imply that the fitness effects of new mutations are mostly conserved across mammalian orthologs, in agreement with other studies showing that for conserved orthologs with similar structures and functions, models without epistasis provide a reasonable estimate of fitness effects in protein-coding genes [67, 68]. Conceptually, the framework presented here, with the addition of a more complex protein fitness landscape at the phylogenetic scale, could be used to infer the relative contribution of compensatory mutations to non-adaptive and adaptive evolution.

## 3.6 Detecting adaptation above the nearly-neutral background

A long-standing debate in molecular evolution is whether the variation we observe between species in protein-coding genes is primarily due to nearly-neutral mutations reaching fixation by drift or primarily due to adaptation [15, 69–71]. Measuring the "rate of adaptation" in proteins, as pioneered by McDonald & Kreitman [5], has been central to inform this debate [72]. However, the McDonald & Kreitman test detects signatures of accelerated evolution in a given terminal branch compared to an expectation based on polymorphism present in the population. It considers the fraction of substitutions that fix too quickly as "adaptive" [5–7] despite there being other processes that can lead to their fixation [73–75] and some of these substitutions being beneficial but non-adaptive [12–14, 17]. Here, the expectation is built on the pattern of substitutions across a phylogeny compared to the fitness effects that can be estimated from both substitutions in a terminal lineage and polymorphism in populations. Moreover, the goal is not to detect a fraction of beneficial substitution (i.e. "adaptive" substitutions for McDonald & Kreitman [5]), but to estimate the proportion of non-adaptive mutations among beneficial ones.

We provide evidence that in mammalian orthologs, many substitutions occur through fixation of both deleterious mutations and beneficial non-adaptive mutations. Detecting adaptation above this background of substitutions remains a challenge [69, 76]. Mathematically, the surplus of positive selection due to an externally-driven changing fitness landscape is called *fitness flux*, and requires experimentally measuring the selection coefficient of each mutation in each genetic background. The *fitness flux* can be estimated if either the substitutions history is known [13] or changes of frequency in currently segregating variants [51]. Without experimentally measured selection coefficients, another strategy is precisely to use a nearly-neutral substitution model as a null model of evolution. Under a strictly neutral evolution of protein-coding sequence, we

expect the ratio of non-synonymous over synonymous substitutions ($d_N/d_S$) to be equal to one. Deviations from this neutral expectation, such as $d_N/d_S > 1$, which can be generated by an excess of non-synonymous substitutions, is generally interpreted as a sign of adaptation. However, as shown in this study, a $d_N/d_S > 1$ is not necessarily a signature of adaptation but can be due to beneficial non-adaptive mutations. So, by relaxing the strict neutrality and assuming a stable fitness landscape instead, one can predict the expected rate of evolution, called $\omega_0$ [77, 78]. Adaptation can thus be considered as evolution under a changing fitness landscape and tested as such by searching for the signature of $d_N/d_S > \omega_0$ [19, 30, 79]. Using a stable fitness landscape as a null model of evolution, thus accounting for selective constraints exerted on the different amino acids, increased the statistical power in testing for adaptation [46]. Instead of relying solely on summary statistics (such as $d_N/d_S$ or $\omega_0$), another strategy to detect adaptation is to include changes in the fitness landscapes inherently within the mutation selection framework, either with small changes along the phylogeny [80] or either by allowing fitness to change on subsets of branches [81, 82]. Such mechanistic models could be more general than site-specific fitness landscapes, including epistasis and changing fitness landscapes [62, 82].

### 3.7 Conclusions

We have provided empirical evidence that an evolutionary model assuming a stable fitness landscape at the mammalian scale allows us to predict the fitness effects of mutations in extant populations and individuals, acknowledging the balance between deleterious and beneficial non-adaptive mutations. We argue that such a model would represent a null expectation for the evolution of protein-coding genes in the absence of adaptation. Altogether, because a substantial part of positive selection can be explained by beneficial non-adaptive mutations, but not its entirety, we argue that the mammalian exome is shaped by both adaptive and non-adaptive processes, and that none of them alone is sufficient to explain the observed patterns of changes. In that sense, to avoid conflating beneficial mutations with adaptive evolution, the term "adaptation" should retain its original meaning associated with a change in the underlying fitness landscape and be modelled as such [13, 51].

## 4 Materials & methods

### 4.1 Phylogenetic dataset

Protein-coding DNA sequence alignments in placental mammals and their corresponding gene trees come from the OrthoMaM database (https://www.orthomam.univ-montp2.fr) and were processed as in Latrille *et al.* [46]. OrthoMaM contains a total of 116 mammalian reference sequences in v10c [33, 34, 83].

Genes located on the X and Y chromosomes and on the mitochondrial genome were discarded from the analysis because the level of polymorphism—which is necessary for population-based analyses—is expected to be different in these three regions compared to the autosomal genome. Sequences of species for which we used population-level polymorphism (see section 4.3) and their sister species, were removed from the analysis to ensure independence between the data used in the phylogenetic and population scales. Sites in the alignment containing more than 10% of gaps across the species were discarded. Altogether, our genome-wide dataset contains 14, 509 protein-coding DNA sequences in 87 placental mammals.

### 4.2 Selection coefficient ($S_0$) in a phylogeny-based method

We analyzed the phylogenetic-level data using mutation-selection models. These models assume the protein-coding sequences are at mutation-selection balance under a fixed fitness

landscape characterized by a fitness vector over the 20 amino acids at each site [26, 28, 84]. Mathematically, the rate of non-synonymous substitution from codon $a$ to codon $b$ ($q_{a \mapsto b}^{(i)}$) at site $i$ of the sequence is equal to the rate of mutation of the underlying nucleotide change ($\mu_{a \mapsto b}$) multiplied by the scaled probability of mutation fixation ($\mathbb{P}_{a \mapsto b}^{(i)}$). The probability of fixation depends on the difference between the scaled fitness of the amino acid encoded by the mutated codon ($F_b^{(i)}$) and the amino acid encoded by the original codon ($F_a^{(i)}$) at site $i$ [85, 86].

The rate of substitution from codon $a$ to $b$ at a site $i$ is thus:

$$\begin{cases} q_{a \mapsto b}^{(i)} & = 0 \text{ if codons } a \text{ and } b \text{ are more than one mutation away,} \\ q_{a \mapsto b}^{(i)} & = \mu_{a \mapsto b} \text{ if codons } a \text{ and } b \text{ are synonymous, and} \\ q_{a \mapsto b}^{(i)} & = \mu_{a \mapsto b} \dfrac{F_b^{(i)} - F_a^{(i)}}{1 - e^{F_a^{(i)} - F_b^{(i)}}} \text{ if codons } a \text{ and } b \text{ are non} - \text{synonymous.} \end{cases} \quad (1)$$

Fitting the mutation-selection model on a multi-species sequence alignment leads to an estimation of the gene-wide $4 \times 4$ nucleotide mutation rate matrix ($\boldsymbol{\mu}$) as well as the 20 amino-acid fitness landscape ($\boldsymbol{F}^{(i)}$) at each site $i$. The priors and full configuration of the model are given in S1 File (section 1). From a technical perspective, the Bayesian estimation is a two-step procedure [87]. The first step is a data augmentation of the alignment, consisting in sampling a detailed substitution history along the phylogenetic tree for each site, given the current value of the model parameters. In the second step, the parameters of the model can then be directly updated by a Gibbs sampling procedure, conditional on the current substitution history. Alternating between these two sampling steps yields a Markov chain Monte-Carlo (MCMC) procedure whose equilibrium distribution is the posterior probability density of interest [87, 88]. Additionally, across-site heterogeneities in amino-acid fitness profiles are captured by a Dirichlet process. More precisely, the number of amino-acid fitness profiles estimated is lower than the number of sites in the alignment. Consequently each profile has several sites assigned to it, resulting in a particular configuration of the Dirichlet process. Conversely, sites with similar signatures are assigned to the same fitness profile. This configuration of the Dirichlet process is resampled through the MCMC to estimate a posterior distribution of amino acid profiles for each site specifically [35, 89]. From a more mechanistic perspective, even though not all amino acids occur at every single codon site of the DNA alignment, we can nevertheless estimate the distribution of amino-acid fitnesses by generalizing the information recovered across sites and across amino acids based on the phylogenetic relationship among samples. In particular, synonymous substitutions along the tree contain the signal to estimate branch lengths and the nucleotide transition matrix, while non-synonymous substitutions contain information on fitness difference between codons connected by single nucleotide changes [35].

The selection coefficient for a mutation from codon $a$ to codon $b$ at site $i$ is defined as:

$$S_0^{(i)}(a \mapsto b) = \Delta F^{(i)} = F_b^{(i)} - F_a^{(i)}. \quad (2)$$

In our subsequent derivation the source ($a$) and target ($b$) codons as well as the site ($i$) are implicit and thus never explicitly written.

The scaled selection coefficient ($S_0 = \Delta F$) is formally the product of the selection coefficient at the individual level ($s$) and the effective population size ($N_e$), as $S_0 = 4N_e \times s$. The value of $S_0$ informs us on the strength of selection exerted on amino acids changes. Thus, according to its $S_0$ value, we can classify any mutation as either a deleterious mutation toward a less fit amino acid ($\mathcal{D}_0 := S_0 < -1$), a nearly-neutral mutation ($\mathcal{N}_0 := -1 < S_0 < 1$), or a mutation toward

a known fitter amino acid, constituting thus a beneficial non-adaptive mutation
($\mathcal{B}_0 \coloneqq S_0 > 1$).

We used the Bayesian software *BayesCode* (https://github.com/ThibaultLatrille/bayescode, v1.3.1) to estimate the selection coefficients for each protein-coding gene in the mammalian dataset. We ran the MCMC algorithm implemented in BayesCode for 2,000 generations as described in Latrille *et al.* [46]. For each gene, after discarding a burn-in period of 1,000 generations of MCMC, we obtained posterior mean estimates (over the 1,000 generations left of MCMC) of the mutation rate matrix ($\boldsymbol{\mu}$) as well as the 20 amino-acid fitness landscape ($\boldsymbol{F}^{(i)}$) at each site *i*.

### 4.3 Polymorphism dataset

The genetic variants representing the population level polymorphisms were obtained from the following species and their available datasets: *Equus caballus* (EquCab2 assembly in the EVA study PRJEB9799 [90]), *Bos taurus* (UMD3.1 assembly in the NextGen project: https://projects.ensembl.org/nextgen/), *Ovis aries* (Oar_v3.1 assembly in the NextGen project), *Capra hircus* (CHIR1 assembly in the NextGen project, converted to ARS1 assembly with dbSNP identifiers [91]), *Chlorocebus sabaeus* (ChlSab1.1 assembly in the EVA project PRJEB22989 [92]), *Homo sapiens* (GRCh38 assembly in the 1000 Genomes Project [93]). In total, we analyzed 28 populations across the 6 different species with polymorphism data. The data was processed as described in Latrille *et al.* [46].

Only bi-allelic single nucleotide polymorphisms (SNPs) found within a gene were in our polymorphism dataset, while nonsense variants and indels were discarded. To construct the dataset, we first recovered the location of each SNP (represented by its chromosome, position, and strand) in the focal species and matched it to its corresponding position in the coding sequence (CDS) using gene annotation files (GTF format) downloaded from Ensembl (ensembl.org). We then verified that the SNP downloaded from Ensembl matched the reference in the CDS in FASTA format. Next, the position in the CDS was converted to the corresponding position in the multi-species sequence alignment (containing gaps) from the OrthoMaM database (see section 4.2) for the corresponding gene by doing a global pairwise alignment (Biopython function pairwise2). This conversion from genomic position to alignment position was only possible when the assembly used for SNP-calling was the same as the one used in the OrthoMaM alignment, the GTF annotations, and the FASTA sequences. SNPs were polarized using the three closest outgroups found in the OrthoMaM alignment with est-usfs v2.04 [94], and alleles with a probability of being derived lower than 0.99 were discarded.

### 4.4 Mutational opportunities

The mutational opportunities of any new mutation refer to its likelihood of falling into a specific category (synonymous, deleterious, nearly-neutral, or beneficial). Deriving such opportunities is necessary to estimate the strength of selection exerted at the population scale since different categories might have different mutational opportunities, and thus polymorphism and divergence need to be corrected accordingly (see sections 4.5, 4.6, and 4.7). To calculate mutational opportunities, we reconstructed the ancestral exome of each of the 28 populations by using the most likely ancestral state from est-usfs (see section 4.3), which differs from the corresponding species reference exome since it accounts for the variability present in the specific population.

From the reconstructed ancestral exome, all possible mutations were computed, weighted by the instantaneous rate of change between nucleotides obtained from the mutation rate matrix ($\boldsymbol{\mu}$, see section 4.2), summing to $\mu_{\mathrm{tot}}$ across the whole exome, and to $\mu_{\mathrm{syn}}$ when restricted

to synonymous mutations. Finally, the mutational opportunities for synonymous mutations were computed as the total number of sites across the exome ($L_{\text{tot}}$) weighted by the proportion of synonymous mutations among all possible mutations as:

$$L_{\text{syn}} = L_{\text{tot}} \frac{\mu_{\text{syn}}}{\mu_{\text{tot}}}. \tag{3}$$

Similarly, for non-synonymous mutations, the total mutation rate for each class of selection $x \in \{\mathcal{D}_0, \mathcal{N}_0, \mathcal{B}_0\}$, called $\mu(x)$, was estimated as the sum across all non-synonymous mutations if their selection coefficient at the phylogenetic scale is in the class $S_0 \in x$. Accordingly, the mutational opportunities ($L(x)$) for each class of selection coefficient ($x$) was finally computed as the total number of sites across the exome ($L_{\text{tot}}$) weighted by the ratio of the aggregated mutations rates falling in the class $\mu(x)$:

$$L(x) = L_{\text{tot}} \frac{\mu(x)}{\mu_{\text{tot}}}. \tag{4}$$

Finally, $\mathbb{P}[x]$ is the probability for a non-synonymous mutation to be in the class $x$, thus computed as:

$$\mathbb{P}[x] = \frac{L(x)}{\sum_{y \in \{\mathcal{D}_0, \mathcal{N}_0, \mathcal{B}_0\}} L(y)}. \tag{5}$$

## 4.5 Substitution mapping and $d_N/d_S$ in the terminal branch

We inferred the protein-coding DNA sequences for each node of the 4-taxa tree containing the focal species and the three closest outgroups species found in the OrthoMaM alignment by applying the M5 codon model (gamma site rate variation) as implemented in FastML.v3.11 [95]. Consequently, for each focal species we reconstructed the protein coding DNA sequence of the whole exome at the base of the terminal branch before the split from the sister species. We considered *Ceratotherium simum simum* as *Equus caballus'* sister species; *Bison bison bison* as *Bos taurus'* sister species; *Pantholops hodgsonii* as *Ovis aries'* sister species; *Pantholops hodgsonii* as *Capra hircus'* sister species; *Macaca mulatta* as *Chlorocebus sabaeus'* sister species and finally, we considered *Pan troglodytes* as *Homo sapiens'* sister species. From this reconstructed exome, we determined the direction of the substitution occurring along the terminal branch of the phylogenetic tree toward each extant population. SNPs segregating in the population were discarded, and the most likely ancestral state from est-usfs (see section 4.3) was used as the reference for each extant population. For each substitution, we recovered its $S_0$ value as calculated through the phylogeny-based method (see section 4.2). Finally, the rate of non-synonymous over synonymous substitutions for a given class of selection coefficient ($x \in \{\mathcal{D}_0, \mathcal{N}_0, \mathcal{B}_0\}$) was computed as:

$$\begin{cases} d_N(x) & = \dfrac{D(x)}{L(x)}, \\[2ex] d_S & = \dfrac{D_{\text{syn}}}{L_{\text{syn}}}, \end{cases} \tag{6}$$

where $D(x)$ was the number of non-synonymous substitutions in class $x$, $D_{\text{syn}}$ was the number of synonymous substitutions across the exome, while $L(x)$ and $L_{\text{syn}}$ were the numbers of non-synonymous and synonymous mutational opportunities, respectively, as defined in section

4.4. $\delta(d_N/d_S)$ was computed as the difference between $d_N/d_S$ computed over all substitutions and $d_N/d_S$ when we removed beneficial non-adaptive mutations $d_N(S_0 < 1)/d_S$, normalized by $d_N/d_S$. Note that the quantities $\delta(d_N/d_S)$ and $\delta(d_N)$ are equivalent due to the simplification of the factor $d_S$:

$$\delta(d_N/d_S) = \frac{d_N/d_S - d_N(S_0 < 1)/d_S}{d_N/d_S} = \frac{d_N - d_N(S_0 < 1)}{d_N} = \delta(d_N). \tag{7}$$

## 4.6 Scaled selection coefficients ($S$) in a population-based method

To obtain a quantitative estimate of the distribution of selection coefficients for each category of SNPs, we used the *polyDFE* model [42, 96]. This model uses the count of derived alleles to infer the distribution of fitness effects (DFE). The probability of sampling an allele at a given frequency (before fixation or extinction) is informative of its scaled selection coefficient at the population scale ($S$). Therefore, pooled across many sites, the site-frequency spectrum (SFS) provides information on the underlying $S$ of mutations. However, estimating a single $S$ for all sampled mutations is biologically unrealistic, and a DFE of mutations is usually assumed [39, 40]. The *polyDFE* [42, 96] software implements a mixture of a $\Gamma$ and exponential distributions to model the DFE of non-synonymous mutations, while synonymous mutations are considered neutral. The model estimates the parameters $\beta_d$, $b$, $p_b$ and $\beta_b$ for non-synonymous mutations as:

$$\phi(S; \beta_d, b, p_b, \beta_b) = \begin{cases} (1 - p_b)f_\Gamma(-S; -\beta_d, b) & \text{if } S \leq 0, \\ p_b f_e(S; \beta_b) & \text{if } S > 0, \end{cases} \tag{8}$$

where $\beta_d \leq -1$ is the estimated mean of the DFE for $S \leq 0$; $b \geq 0.2$ is the estimated shape of the $\Gamma$ distribution; $0 \leq p_b \leq 1$ is the estimated probability that $S > 0$; $\beta_b \geq 1$ is the estimated mean of the DFE for $S > 0$; and $f_\Gamma(S; m, b)$ is the density of the $\Gamma$ distribution with mean $m$ and shape $b$, while $f_e(S; m)$ is the density of the exponential distribution with mean $m$.

*PolyDFE* requires one SFS for non-synonymous mutations and one for synonymous mutations (neutral expectation), as well as the number of sites on which each SFS was sampled. For populations containing more than 8 individuals, the SFS was subsampled down to 16 chromosomes (8 diploid individuals) without replacement (hyper-geometric distribution) to alleviate the effect of different sampling depths in the 28 populations. Altogether, for each class of selection ($x \in \{\mathcal{D}_0, \mathcal{N}_0, \mathcal{B}_0\}$) of non-synonymous SNPs, we aggregated all the SNPs in the selection class $x$ as an SFS. The number of sites on which each SFS was sampled is given by $L(x)$ for the non-synonymous SFS and $L_{\text{syn}}$ for the synonymous SFS respectively. For each class of selection $x$, once fitted to the data using maximum likelihood with *polyDFE*, the parameters of the DFE ($\beta_d$, $b$, $p_b$, $\beta_b$) were used to compute $\mathbb{P}[\mathcal{D} \mid x]$, $\mathbb{P}[\mathcal{N} \mid x]$, and $\mathbb{P}[\mathcal{B} \mid x]$ as:

$$\mathbb{P}[\mathcal{D} \mid x] = \mathbb{P}[S < -1 \mid x] = (1 - p_b) \int_{-\infty}^{-1} f_\Gamma(-S; -\beta_d, b)\mathrm{d}S, \tag{9}$$

$$\mathbb{P}[\mathcal{N} \mid x] = \mathbb{P}[-1 < S < 1 \mid x] = (1 - p_b) \int_{-1}^{0} f_\Gamma(-S; -\beta_d, b)\mathrm{d}S + p_b \int_{0}^{1} f_e(S; \beta_b)\mathrm{d}S, \tag{10}$$

$$\mathbb{P}[\mathcal{B} \mid x] = \mathbb{P}[S > 1 \mid x] = p_b \int_1^{+\infty} f_e(S; \beta_b)\mathrm{d}S. \tag{11}$$

Rather than relying solely on currently segregating mutations to quantify selection, *polyDFE* can leverage both divergence and polymorphism to estimate the parameters of the DFE. We can thus add four more inputs to *polyDFE*: $D(x)$, $L(x)$, $D_{\mathrm{syn}}$ and $L_{\mathrm{syn}}$ such as defined in the previous section. Because the estimates of DFE are different with this method, we naturally obtained different values of $\mathbb{P}[\mathcal{D} \mid x]$, $\mathbb{P}[\mathcal{N} \mid x]$, and $\mathbb{P}[\mathcal{B} \mid x]$.

### 4.7 *Precision* and *recall*

For readability, we give here *precision* and *recall* for beneficial mutations ($\mathcal{B}_0$ and $\mathcal{B}$), but it can be obtained using the same derivation for the deleterious mutations ($\mathcal{D}_0$ and $\mathcal{D}$) and nearly-neutral mutations ($\mathcal{N}_0$ and $\mathcal{N}$).

*Precision* is the proportion of mutations correctly predicted as beneficial ($\mathbb{P}[\mathcal{B} \cap \mathcal{B}_0]$) out of all predicted as beneficial non-adaptive mutations ($\mathbb{P}[\mathcal{B}_0]$), which can be written as a conditional probability:

$$\frac{\mathbb{P}[\mathcal{B} \cap \mathcal{B}_0]}{\mathbb{P}[\mathcal{B}_0]} = \mathbb{P}[\mathcal{B} \mid \mathcal{B}_0]. \tag{12}$$

Namely, *precision* corresponds to the probability for a $\mathcal{B}_0$ mutation to be effectively beneficial at the population level ($\mathcal{B}$). This probability, computed from Eq 11, is obtained by restricting our analysis to SNPs that are predicted to be beneficial non-adaptive mutations (yellow fill for the category $\mathcal{B}_0$ in Fig 3D).

*Recall* is the proportion of mutations correctly predicted as beneficial ($\mathbb{P}[\mathcal{B} \cap \mathcal{B}_0]$) out of all beneficial mutations ($\mathbb{P}[\mathcal{B}]$), which can be written as a conditional probability:

$$\frac{\mathbb{P}[\mathcal{B} \cap \mathcal{B}_0]}{\mathbb{P}[\mathcal{B}]} = \mathbb{P}[\mathcal{B}_0 \mid \mathcal{B}]. \tag{13}$$

Namely, *recall* corresponds to the probability for a beneficial mutation at the population level ($\mathcal{B}$) to be a beneficial non-adaptive mutation ($\mathcal{B}_0$). Using Bayes theorem, *recall* can be re-written as:

$$\mathbb{P}[\mathcal{B}_0 \mid \mathcal{B}] = \frac{\mathbb{P}[\mathcal{B} \mid \mathcal{B}_0] \times \mathbb{P}[\mathcal{B}_0]}{\mathbb{P}[\mathcal{B}]}, \tag{14}$$

where $\mathbb{P}[\mathcal{B} \mid \mathcal{B}_0]$ and $\mathbb{P}[\mathcal{B}_0]$ can be calculated using Eqs 12 and 5, respectively, and $\mathbb{P}[\mathcal{B}]$ is the probability of a mutation to be beneficial at the level of the population, which can be computed from the law of total probabilities as:

$$\mathbb{P}[\mathcal{B}] = \sum_{x \in \{\mathcal{D}_0, \mathcal{N}_0, \mathcal{B}_0\}} \mathbb{P}[\mathcal{B} \mid x] \times \mathbb{P}[x]. \tag{15}$$

### 4.8 Correlation with effective population size ($N_e$)

Genetic diversity estimator Watterson's $\theta_S$ was obtained for each population from the synonymous SFS as in Achaz [45]. For each popuation, $N_e$ was estimated from the equation $N_e = \theta_S/(4 \times u)$, where $u$ is the mutation rate per generation. Estimates for $u$ were averaged per species

across the pedigree-based estimation in Bergeron et al. [43] for *Homo*, *Bos*, *Capra* and *Chlorocebus*. For *Ovis* we used the estimated *u* of *Capra*. For *Equus*, we used *u* as estimated in Orlando *et al.* [44] ($u = 7.24 \times 10^{-9}$). Because a correlation must account for phylogenetic relationship and non-independence of samples, we fitted a Phylogenetic Generalized Linear Model in *R* with the method `pgls` with default settings from the package `caper` [97]. The mammalian dated tree was obtained from TimeTree [98] and pruned to include only the species analysed in this study, with multi-furcation of the different populations from each species placed at the same divergence time as the species (section 2.1 in S3 File).

## Supporting information

**S1 File. Supplementary appendix on the parameterization of Mutation-selection codon models.** Contains 5 pages of supplementary information including 1 figure (Fig A).
(PDF)

**S2 File. Supplementary appendix on the caracterization of non-adaptive beneficial mutations.** Contains 12 pages of supplementary information including 3 figures (Fig A to C) and 6 tables (Table A to F).
(PDF)

**S3 File. Supplementary appendix on the contrast of selection at the phylogenetic and population-genetic scales.** Contains 7 pages of supplementary information including 8 figures (Fig A to H).
(PDF)

**S4 File. Supplementary appendix on controls for estimating the proportion of beneficial mutations that are not adaptive.** Contains 21 pages of supplementary information including 9 figures (Fig A to I) and 9 tables (Table A to I).
(PDF)

## Acknowledgments

We gratefully acknowledge the help of Mélodie Bastian, Nicolas Lartillot, Carina Farah Mugal, Laurent Duret, Alexandre Reymond, Daniele Silvestro and Nicolas Gambardella for their advice and reviews concerning this manuscript. This work was performed using the computing facilities of the CC LBBE/PRABI. This study makes use of data generated by the NextGen Consortium.

## Author Contributions

**Conceptualization:** Thibault Latrille, Julien Joseph, Nicolas Salamin.

**Data curation:** Thibault Latrille, Julien Joseph.

**Formal analysis:** Thibault Latrille, Julien Joseph.

**Funding acquisition:** Nicolas Salamin.

**Investigation:** Thibault Latrille, Julien Joseph, Diego A. Hartasánchez.

**Methodology:** Thibault Latrille, Julien Joseph.

**Project administration:** Nicolas Salamin.

**Resources:** Thibault Latrille, Nicolas Salamin.

**Software:** Thibault Latrille.

**Supervision:** Nicolas Salamin.

**Validation:** Thibault Latrille, Julien Joseph, Diego A. Hartasánchez, Nicolas Salamin.

**Visualization:** Thibault Latrille, Julien Joseph, Diego A. Hartasánchez.

**Writing – original draft:** Thibault Latrille, Julien Joseph, Diego A. Hartasánchez.

**Writing – review & editing:** Thibault Latrille, Julien Joseph, Diego A. Hartasánchez, Nicolas Salamin.

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
