## [Decision Letter · Decision Letter 0]

13 Jun 2024

Dear Dr Latrille,

Thank you very much for submitting your Research Article entitled 'Estimating the proportion of beneficial mutations that are not adaptive in mammals' to PLOS Genetics.

The manuscript was fully evaluated at the editorial level and by independent peer reviewers. The reviewers appreciated the attention to an important problem, but raised some substantial concerns about the current manuscript. Based on the reviews, we will not be able to accept this version of the manuscript, but we would be willing to review a much-revised version. We cannot, of course, promise publication at that time.

If you decide to revise the manuscript for further consideration at PLOS Genetics, please aim to resubmit within the next 60 days, unless it will take extra time to address the concerns of the reviewers, in which case we would appreciate an expected resubmission date by email to plosgenetics@plos.org.

We are sorry that we cannot be more positive about your manuscript at this stage. Please do not hesitate to contact us if you have any concerns or questions.

Yours sincerely,

Kirk E Lohmueller

Guest Editor

PLOS Genetics

Justin Fay

Section Editor

PLOS Genetics

If you choose to submit a revised manuscript, please address all of the comments from the reviewers. When revising your manuscript, please pay special attention to the following points:

1) Please add some simulations to address the performance of your approach under the assumptions of the model.

2) Reviewer 3 raises concerns about the functional form for the DFE that you’re fitting. Please carefully address these points.

3) Reviewer 2 asks questions pertaining to the role that changing fitness landscapes could play in the inference. Reviewer 2 also raises concerns about the correlation of the patterns with effective population size. Please resolve this point.

Reviewer's Responses to Questions

**Comments to the Authors:**

Reviewer #1: The authors provides a very careful evaluation of the fitness effect of non-synonymous mutations using both population genetic and phylogenetic (multi-species alignment) data and methods. They suggest that a substantial proportion (between 15-45%) of all new beneficial mutations are “non-adaptive”, i.e. they restore fitness that was lost due to a previous fixation of a slightly deleterious mutation and thus are not responding to a changed environment. This is as far as I know the first quantitative assessment of this idea. A central assumption is that the fitness landscape of amino acid replacements can be estimated from phylogenetic data and that this fitness landscape stays constant over time. There are some potential model violations (changing population size, epistasis, CpG mutation bias) and the authors provide a fair assessment each of these potential issues and how these could affect the precision of the estimated effect.

I don’t have any major concerns. This is a careful and insightful study and I fully endorse publication in PLOS Genetics. Below, I’ve listed a few suggestions for the discussion.

I think it would be useful to have a more explicit discussion of strongly beneficial mutations. Rare but strongly beneficial mutations would not be inferred by either of the methods, but can contribute substantially to divergence. I think it would help to mention that the inferred beneficial (non-adaptive) mutations are relatively weak and fall within a narrow range of selection strength (1< S <10). I also wonder if it is possible to calculate how much non-adaptive beneficial mutations contribute to nonsynonymous divergence, relative to all beneficial mutations. E.g., how much would they inflate estimates of the rate of adaptive evolution (alpha)?

Related to this, could the authors comment on the study by Endard et al. (eLife, 2016, https://doi.org/10.7554/eLife.12469), which states that viruses are dominantly driving protein adaptation in mammals? They estimate that close to 30% of adaptive amino acid changes are driven by adaptation to viruses. Is this consistent with or contradicting the results of the present study?

Reviewer #2: In this paper, Latrille, Joseph et al. combine phylogenetic and popopulation genetic methods to infer the contribution of beneficial non-adaptive mutations to molecular evolution. They define beneficial non-adaptive mutations as mutations that revert deleterious amino acid changes that have fixed due to drift to their previous, beneficial state. This is contrasted with adaptive mutations, which increase in frequency due to positive selection after a change in the fitness landscape, compensatory mutations, which restore ancestral fitness through a change at a different amino acid than the initial deleterious change, and nearly neutral mutations, which have little fitness effect. They present evidence that ongoing positive selection is driven by a combination of beneficial non-adaptive mutations as well as adaptive mutations, and the proportion of beneficial non-adaptive mutations may be substantial. In addition, they make the point that a model that includes positively-selected beneficial non-adaptive mutations is a more appropriate null model for the evolution of protein-coding genes than is a model of no positive selection. Overall, this work is an interesting and important contribution to our understanding of adaptation and positive selection.

Major comments

My major comments are centered on two main things: The assumption of a stable underlying fitness landscape, and the consequences of the dependence on effective population size.

Stable underlying fitness landscape:

Overall, I would appreciate more clarification and reasoning for why the assumption of a stable underlying fitness landscape is reasonable in this case. It is necessary for the model employed, which makes it extremely important to both justify its use and consider the consequences if the assumption is violated. In the scenario as formulated, the presence of adaptive mutations means the fitness landscape is not entirely stable (acknowledged by the authors eg. P4L113-114 “we could tease apart beneficial non-adaptive from adaptive mutations resulting from a change in the fitness landscape”). Is the implication then that the fitness landscape is stable for some proteins, but not for others? If so, this could have interesting implications for understanding the relative proportions of beneficial non-adaptive mutations vs. adaptive mutations.

Furthermore, what would be the theoretical consequences if this assumption were relaxed? The authors start to approach this in one way by re-conducting the analysis using highly conserved genes (P7L208-211), which might be expected to have a more stable fitness landscape than other genes that are likely to adapt to changing environmental conditions. I would appreciate more discussion of the implications of the findings associated with that version of the analysis in general, and specifically as they might relate to the assumption of a stable fitness landscape.

In addition, could the presence of (an unknown number of) adaptive mutations impact the fitting of the mutation-selection model to the multi-species alignments, and impact the estimation of the amino-acid fitness landscape itself?

This potential concern is especially pronounced for species that have been undergoing domestication, and as a result have had major changes in their fitness landscapes in the relatively recent past. Four of the six species tested (Ovis, Capra, Bos, Equus) are domestic species, and as a result are expected to have had strong adaptive responses to a major shift in their fitness landscape, potential changes in the DFE as a result (Rice et al. 2015 Genetics), and additionally recent large reductions in effective population size.

Effective population size:

Because the definition of mutation categories deleterious, nearly-neutral, and (non-adaptive) beneficial is relative to S0, and S0 = 4Ne*s, the categorization of mutations necessarily depends on species/population Ne. This is appropriate, since efficiency of selection depends on Ne, but it means that the Ne estimates are very important to the results. I appreciate that the authors took a consistent approach based on θ, but their reported Ne values are 1-2 orders of magnitude higher for many of these species than often reported in the literature (for example, see Warren et al. 2015 for Chlorocebus sabaeus; Eyidivandi et al. 2020 Animal Genetics, Taheri et al. 2022 Small Ruminant Research for Ovis aries spp. Taheri et al. also used the NextGen data.) If the Nes reported here are biased upward, that will also bias the proportion of mutations determined to be in each category.

In addition, phylogenetic regressions are conducted to understand the relationship between the proportion of mutations in each category at the population scale and at the population scale given its category at the phylogenetic scale. However the expectations of these regressions are not straightforward, since Prop[B], Prop[B0|B], and the proportions of the other categories are, by definition, dependent on Ne. As a result, the expected relationships are dependent on Nes. That is, a relationship between the dependent variable and Ne is baked in.

Minor comments

- The structure of the manuscript is at times hard to follow, with material in the introduction that seems to fit better in the methods, etc. I ask the authors to revisit the overall structure during their revisions and make sure that the organization is straightforward. Some examples:

- Some methodological descriptions in introduction rather than methods, these make it hard to follow/find these details (eg. P3L90-98)

- P8L250-251 should be in results

- P16L486-487 “we fitted a Phylogenetic Generalized Linear Model in R with the package caper” it is unclear without more context (found only in the results and discussion) what the nature of this model is.

- Eqs. 9 and 10 are inconsistent with eq. 8, showing fΓ(S; -βd,b) vs. fΓ(-S; -βd,b) Is this intentional?

- The authors bring up compensatory mutations a couple of times throughout the manuscript, including in a section of the discussion in which they discuss the potential conflation of putatively adaptive mutations and compensatory mutations in their study. They argue that the proportion of beneficial non-adaptive mutations observed in the present study supports that a model without epistasis as an adequate fit to the protein data (P8 L277-278). I would appreciate additional clarification of this argument, as on the surface it seems like a high proportion of observed beneficial non-adaptive mutations wouldn’t preclude an additional contribution of compensatory mutations. In addition, this makes me curious about whether the method presented, with the addition of a more complex protein fitness landscape, could be used to infer the relative contribution of compensatory mutations (with perhaps further implications for inferring the contribution of compensatory mutations to adaptive evolution, see eg. Storz 2018 “Compensatory mutations and epistasis for protein function”)?

- The authors point out the difficulty in distinguishing between beneficial non-adaptive mutations from convergent adaptation, which is an interesting point. I wonder if incorporating effective population size of the lineages that include such convergent/beneficial non-adaptive mutations could help to distinguish them, given the expected relationship between the proportion of mutations of different classes and effective population size?

Reviewer #3: The paper concerns the prevalence of advantageous back mutations in molecular evolution. The authors present a method for estimating this prevalence. The method is based on fitting a model of molecular evolution where each site in a protein has a conserved set of amino acid preferences given as scaled fitness values. Then, currently segregating SNPs can be assigned scaled selection coefficients under this model. The site frequency spectrum of these same SNPs can be used to estimate the distribution of current selective pressures as a function of the predicted selection coefficients from the model of long-term amino acid preferences. This can then be used to estimate the fraction of beneficial new mutations that are restoring a historically preferred amino acid state versus the fraction of beneficial new mutations that are responding to new selective pressures. The authors apply the method exome-wide to a set of 80+ mammalian species for the model of molecular evolution and polymorphisms from over 20 populations. While the estimates for the prevalence of advantageous back mutations vary widely between populations, they generally constitute a substantial minority of all beneficial mutations.

The manuscript represents an innovative way of combining approaches from molecular evolution and population genetics to address an interesting but understudied problem (the prevalence of beneficial back mutations). The broad strokes approach is similar to a recent PNAS paper with the same lead author (Latrille, T., Rodrigue, N. & Lartillot, N. 2023 Genes and Sites under Adaptation at the Phylogenetic Scale Also Exhibit Adaptation at the Population-Genetic Scale. PNAS), but the question and methods here are distinct. The manuscript is a good fit for the readership of PLOS Genetics and the conclusions are of broad interest. That being said, I have some technical concerns about whether the proposed method is sound as currently implemented.

Major issues:

1. While the authors do a good job anticipating the effects of violations of model assumptions (which would tend to result in an underestimate of the frequency of beneficial back mutations), they do not show that their method is capable of producing accurate estimates when the model assumptions are satisfied. They should show this via simulations where the ground truth is known.

2. I am particularly concerned about the previous point because I suspect that the method has a problem as currently implemented. The problem is that the modeling of the DFE of the segregating variants is insufficiently expressive for the role that it plays in the inference. The DFE fit by the authors is a reasonable choice for the overall DFE but they are asking it to fit distributions that may look very different than the overall DFE (e.g. distributions truncated at -1 or +1). Fitting this grossly misspecified model will lead to highly inaccurate parameter estimates in some regimes.

Specifically, the DFE they use is a mixture between a reflected gamma distribution and an exponential distribution, which is a standard choice in the literature for estimating the whole DFE, and is roughly appropriate in that it can e.g. produce the peak at neutrality observed in nature and control the fractions of slightly deleterious mutations, etc. However, under the authors’ procedure they split the SNPs into subsets based on their estimated selection coefficients from the phylogenetic model and then estimate the DFEs for these subsets individually. These DFEs will have a completely different shape from the overall DFE. For example, in the case that selective pressures are truly constant over time, the ground truth of the DFEs to be estimated would be a deleterious distribution right-truncated at -1, a neutral distribution truncated between -1 and +1, and a beneficial distribution truncated on the left at +1. Even to the extent that the authors’ model of the DFE can put the right total probability in any one of these regions, the shape of the distribution in those regions will be completely wrong. For example, attempting to put most of the probability greater than +1 will also necessarily produce a DFE with enormously large selection coefficients because the positive portion of the DFE is constrained to be exponential.

Off the top of my head I do not know how to solve this problem. I believe the new fastDFE software from Thomas Bataillon’s group allows you to define custom parametric models for the DFE. Another possibility might be to fit a mixture of the parametric DFE the authors are currently using and the truncated DFEs that would be predicted from the phylogenetic mutation-selection model (but I haven’t thought it through).

3. There are several passages such as line 146-149 stating that dN/dS is “biased” as a measure of adaptation due to the presence of beneficial back mutations. As currently written this discussion appears misguided.

First, it is well-understood that as a practical matter dN/dS is massively biased AGAINST detecting adaptation because the signal from sites under purifying selection typically overwhelms the signal from sites experiencing positive selection. Obviously any discussion of subtle details of the behavior of dN/dS as a method of detecting adaptation has to be viewed against this backdrop.

Second, the authors come to their conclusion by calculating dN/dS for the subset of sites that they predict are maladapted at the start of the terminal branches under their phylogenetic model. Finding dN/dS>1 for these sites is 100% appropriate, since this subset of sites did truly adapt and the new amino acids at these sites really are closer to optimality at the end of the branch than at the beginning! The fact that the cause of the maladaptation is the previous fixation of deleterious mutations rather than environmental change does not change the fact that this subset of sites has contributed positively to the adaptedness of the population. It is just that under the stationary process the adaptation at these sites is offset by maladaptive substitutions elsewhere in the genome so that there is no net adaptation.

Third, there is a broader issue throughout the manuscript of verbally tying adaptation to environmental change. However, rapid evolution in genes with elevated dN/dS is often not driven by environmental change but rather by frequency-dependent selection, e.g. rapid evolution of reproductive genes and genes involved in sexual selection. In the mathematical framework used by the current authors, Mustonen and Lassig 2010 PNAS’s treatment of these issues via the calculation of “fitness flux” provides a rigorous method for addressing these complications precisely in the setting of allelic preferences relevant to the methods described here (which for instance structurally cannot address issues of absolute fitness, reproductive excess, etc.). I think the more intuitive language used in the manuscript is OK as motivation, but the reader should be more explicitly directed to the mathematical theory for a rigorous treatment (and the authors may also want to update some of their views in light of this clearer understanding).

4. The authors do a good job in addressing how epistasis due to interactions between specific sites such as structural contacts would play into their modeling framework but should also address epistasis that arises because mutations act additively but selection is nonlinear (e.g. stabilizing selection on an additive trait such as protein activity). The molecular evolution in the latter case is dealt with by Charlesworth 2013 “Stabilizing Selection, Purifying Selection, and Mutational Bias in Finite Populations” in Genetics.

5. I think there is a missed opportunity in the analysis of substitutions along terminal branches to provide an estimate of the total frequency of amino acid substitutions due to mutation-selection-drift balance. Because the model of molecular evolution used is time reversible, the distribution of mutations fixed at stationarity is symmetric around zero. The lack of symmetry along the terminal branches then provides a means to estimate the frequency of beneficial back substitutions (in contrast to the rest of the study which estimates the frequency of beneficial back mutations). For example, one could use twice the fraction of substitutions with selection coefficients greater than 1 as an estimate of this rate; more refined estimates are also likely possible. Likewise, the fraction of deleterious substitutions minus the fraction of advantageous substitutions provides a crude estimate of the fraction of substitutions due to adaptation to new environmental conditions.

More generally, the highly consistent estimates of the frequency of beneficial back substitutions in the last column of table S1 (very tight between .09-.12) suggests an overall frequency of nearly neutral substitutions due to consistent long-term selection pressures, mutation and drift of approximately 20% of all substitutions (doubling the last column of table S1 to count the corresponding deleterious fixations). This provides additional evidence for the importance of beneficial but not adaptive mutations.

Minor issues:

- Line 167, perhaps also helpful to cite Grimm et al. 2015 here

- I think readers would appreciate a little more discussion on the relationship between these results and the McDonald-Kreitman test literature, since both are integrating fixed and segregating variation, and relate to a fraction of adaptive mutations/substitutions.

- I suggest also discussing the alternative approach of considering adaption to new selective environments by allowing selective preferences to change on subsets of branches (e.g. Kazmi and Rodrigue 2019, or Ritchie, Stark and Liberles 2021) and the strengths / weaknesses of this versus the current approach.

**Have all data underlying the figures and results presented in the manuscript been provided?**

Reviewer #1: Yes

Reviewer #2: Yes

Reviewer #3: Yes

PLOS authors have the option to publish the peer review history of their article (what does this mean?). If published, this will include your full peer review and any attached files.

Reviewer #1: **Yes: **Christian Huber

Reviewer #2: No

Reviewer #3: No

---

## [Decision Letter · Decision Letter 1]

19 Nov 2024

PGENETICS-D-24-00471R1Estimating the proportion of beneficial mutations that are not adaptive in mammalsPLOS Genetics Dear Dr. Latrille, Thank you for submitting your manuscript to PLOS Genetics. After careful consideration, we feel that it has merit but does not fully meet PLOS Genetics's publication criteria as it currently stands. Therefore, we invite you to submit a revised version of the manuscript that addresses the points raised during the review process. Please submit your revised manuscript within 30 days . If you will need more time than this to complete your revisions, please reply to this message or contact the journal office at plosgenetics@plos.org. Please include the following items when submitting your revised manuscript: *
A rebuttal letter that responds to each point raised by the editor and reviewer(s). You should upload this letter as a separate file labeled 'Response to Reviewers'. This file does not need to include responses to formatting updates and technical items listed in the 'Journal Requirements' section below. *
A marked-up copy of your manuscript that highlights changes made to the original version. You should upload this as a separate file labeled 'Revised Manuscript with Track Changes'. *
An unmarked version of your revised paper without tracked changes. You should upload this as a separate file labeled 'Manuscript'. If you would like to make changes to your financial disclosure, competing interests statement, or data availability statement, please make these updates within the submission form at the time of resubmission. Guidelines for resubmitting your figure files are available below the reviewer comments at the end of this letter. We look forward to receiving your revised manuscript. Kind regards,Kirk E LohmuellerAcademic EditorPLOS Genetics Justin FaySection EditorPLOS Genetics Aimée DudleyEditor-in-ChiefPLOS Genetics Anne GorielyEditor-in-ChiefPLOS Genetics **Additional Editor Comments:** Thank you for submitting the revised manuscript. The reviewers and I all agree that it substantially improved. Before we move forward with your manuscript, please address the comments from Reviewer 3 about the Supplementary Information. **Journal Requirements:**

1) Your current Financial Disclosure states, " This work was funded by Université de Lausanne (https://www.unil.ch; to TL, DAH and NS) and Agence Nationale de la Recherche (https://anr.fr/; grant ANR-19-CE12-0019 / HotRec to JJ). This study makes use of data generated by the NextGen Consortium. The European Union’s Seventh Framework Programme (FP7/2010-2014) provided funding for the project under grant agreement no 244356 - “NextGen”. The funders did not play any role in the study design, data collection and analysis, decision to publish, or preparation of the manuscript. " While your funding information on the submission form indicates only two funders. However, the funder "The European Union’s Seventh Framework Programme (FP7/2010-2014)" is currently missing. Please ensure that the funders and grant numbers match between the Financial Disclosure field and the Funding Information tab in your submission form. Note that the funders must be provided in the same order in both places as well. Please indicate by return email the full and correct funding information for your study and confirm the order in which funding contributions should appear. 

**Reviewers' comments:**Reviewer's Responses to Questions

**Comments to the Authors:**

Reviewer #1: The authors have carefully considered and responded to all of my previous comments, and I believe the revisions have enhanced the manuscript.

Reviewer #2: The authors have completed a thorough and substantial review in response to my and the other reviewers’ comments.

I appreciate the additional discussion of predictions at the population vs phylogenetic level and the fitness seascape, as well as the manner in which they addressed the implications of Ne under multiple circumstances. The manuscript is now also easier to follow.

Reviewer #3: The authors have conducted a thorough and thoughtful revision, both with respect to my comments and the comments of the other reviewers, and have conducted numerous additional analyses that bolster the conclusions of the manuscript.

I do suggest some additional minor changes to the SI for formatting and completeness. Specifically, the SI is written as more of an outline than a publication-ready document. For example, it mostly has lists of bullet points below the figures rather than figure captions. Also, I think that many of the supplemental figures could use more explanation and be more self-contained. For instance, I assume that Figure S10 is using the same convention as main text Figure 4 that the circles denote populations and the squares denote species means, but there isn’t a caption and the bulleted text doesn’t explain this either. Likewise, section 7.1 is just a table (plus bullet points acting as a caption) with no explanation as are several of the other SI suggestions. Overall I think the SI should be brought up to a publication-ready standard.

Minor comments:

SI Typo “Additionally to including divergence, with also tested our prediction with polyDFE model D instead of model C” with->we

**Have all data underlying the figures and results presented in the manuscript been provided?**

Reviewer #1: Yes

Reviewer #2: Yes

Reviewer #3: Yes

PLOS authors have the option to publish the peer review history of their article (what does this mean?). If published, this will include your full peer review and any attached files.

Reviewer #1: **Yes: **Christian Huber

Reviewer #2: **Yes: **M. Elise Lauterbur

Reviewer #3: No

**Figure resubmission:** While revising your submission, please upload your figure files to the Preflight Analysis and Conversion Engine (PACE) digital diagnostic tool, https://pacev2.apexcovantage.com/. PACE helps ensure that figures meet PLOS requirements. To use PACE, you must first register as a user. Registration is free. Then, login and navigate to the UPLOAD tab, where you will find detailed instructions on how to use the tool. If you encounter any issues or have any questions when using PACE, please email PLOS at figures@plos.org. Please note that Supporting Information files do not need this step. If there are other versions of figure files still present in your submission file inventory at resubmission, please replace them with the PACE-processed versions. **Reproducibility:** To enhance the reproducibility of your results, we recommend that authors deposit laboratory protocols in protocols.io, where a protocol can be assigned its own identifier (DOI) such that it can be cited independently in the future. Additionally, PLOS ONE offers an option to publish peer-reviewed clinical study protocols. Read more information on sharing protocols at https://plos.org/protocols?utm_medium=editorial-email&utm_source=authorletters&utm_campaign=protocols

---

## [Editor Report · Decision Letter 2]

10 Dec 2024

Dear Dr Latrille,

We are pleased to inform you that your manuscript entitled "Estimating the proportion of beneficial mutations that are not adaptive in mammals" has been editorially accepted for publication in PLOS Genetics. Congratulations!

Yours sincerely,

Kirk E Lohmueller

Academic Editor

PLOS Genetics

Justin Fay

Section Editor

PLOS Genetics

Aimée Dudley

Editor-in-Chief

PLOS Genetics

Anne Goriely

Editor-in-Chief

PLOS Genetics

Comments from the reviewers (if applicable):

**Data Deposition**

http://datadryad.org/submit?journalID=pgenetics&manu=PGENETICS-D-24-00471R2

**Press Queries**

---

## [Editor Report · Acceptance letter]

19 Dec 2024

PGENETICS-D-24-00471R2 

Estimating the proportion of beneficial mutations that are not adaptive in mammals 

Dear Dr Latrille, 

We are pleased to inform you that your manuscript entitled "Estimating the proportion of beneficial mutations that are not adaptive in mammals" has been formally accepted for publication in PLOS Genetics! Your manuscript is now with our production department and you will be notified of the publication date in due course.

With kind regards,

Anita Estes

PLOS Genetics

On behalf of:
